# Gene Expression Profiling of MicroRNAs in HPV-Induced Warts and Normal Skin

**DOI:** 10.3390/biom9120757

**Published:** 2019-11-21

**Authors:** Laith N. AL-Eitan, Mansour A. Alghamdi, Amneh H. Tarkhan, Firas A. Al-Qarqaz

**Affiliations:** 1Department of Applied Biological Sciences, Jordan University of Science and Technology, Irbid 22110, Jordan; amneht92@gmail.com; 2Department of Biotechnology and Genetic Engineering, Jordan University of Science and Technology, Irbid 22110, Jordan; 3Department of Anatomy, College of Medicine, King Khalid University, Abha 61421, Saudi Arabia; m.alghamdi@kku.edu.sa; 4Department of Internal Medicine, Jordan University of Science and Technology, Irbid 22110, Jordan; fqarqaz@just.edu.jo; 5Division of Dermatology, Department of Internal Medicine, King Abdullah University Hospital Jordan University of Science and Technology, Irbid 22110, Jordan

**Keywords:** HPV, wart, microRNA, RNA-seq

## Abstract

Infection with the human papillomavirus (HPV) is a common occurrence among the global population, with millions of new cases emerging on an annual basis. Dysregulated microRNA (miRNA) expression is increasingly being identified to play a role in a number of different diseases, especially in the context of high-risk HPV infection. The present study investigated the miRNA expression profiles of warts induced by low-risk HPV. In warts, miR-27b, miR-24-1, miR-3654, miR-647, and miR-1914 were downregulated while miR-612 was upregulated compared to normal skin. Using miRTargetLink Human, experimentally supported evidence was obtained showing that miR-27b targeted the vascular endothelial growth factor C (*VEGFC*) and CAMP-responsive element binding protein 1 (*CREB1*) genes. The *VEGFC* and *CREB1* genes have been reported to be involved in tumorigenesis and wart formation, respectively. Similarly, the oxidized low-density lipoprotein receptor 1 (*OLR1*) gene, which plays an important role in the humoral immunity of the skin, and the plexin D1 (*PLXND1*) gene, which is highly expressed in tumor vasculature, were both found to be common targets of miR-27b, miR-1914, and miR-612.

## 1. Introduction

Human skin is an integumentary organ that consists of two main layers, the outermost epidermis and the underlying dermis, connected by the basement membrane [1]. The multilayered epidermis is predominantly made up of keratinocyte cells that originate in its basal layer and function to create a physical barrier against environmental and pathogenic threats [2]. Furthermore, keratinocytes faced with invasion by pathogens can induce apoptosis, trigger inflammatory responses, and produce various types of antimicrobial peptides as a means of innate immune defense [3,4,5]. One pathogen that exclusively targets keratinocytes found in the basal epidermal layer is the human papillomavirus (HPV) [6].

As the most common sexually transmitted infection, HPV infects millions of people each year and can be spread by skin-to-skin contact [7]. However, the majority of individuals infected with HPV do not exhibit any clinical symptoms due to quick resolution by the host’s immune system [8,9]. Hundreds of HPV types have been identified and are classified into five genera based on sequence homology of the L1 part of the genome [10]. However, HPV types can also be grouped depending on their risk levels as well as whether they infect the cutaneous or mucosal areas of the skin [11]. High-risk HPV infection often manifests in the anogenital area and results in malignant progression to cervical and other squamous cell carcinomas [12]. By contrast, low-risk HPV types are usually associated with productive skin lesions known as cutaneous warts, which are often benign in nature [13]. The most common type of cutaneous wart, especially in younger children, is the common wart [14,15]. In immunocompetent individuals, warts are oftentimes harmless and self-limited, resolving themselves within a few years if not subjected to any treatment [16].

The HPV replication cycle within the keratinocyte is tightly linked to the differentiation process of the latter, and this regulatory linkage is reflected in the expression profiles of the infected cells [17]. MicroRNAs (miRNAs), a highly conserved subset of noncoding ribonucleic acids, regulate gene expression in a number of contexts, including HPV infection and keratinocyte differentiation [18,19,20]. Although they are thought to make up less than 5% of the human genome, miRNAs modulate the expression of up to 60% of all protein-coding genes [21,22]. In humans, miRNAs function to post-transcriptionally alter gene expression in one of two ways: repression of translation and degradation of messenger RNAs (mRNAs) [23].

Increasing evidence has established that modulation of miRNA expression by high-risk HPV infection contributes to oncogenesis [24,25]. For instance, miR-206 was found to be downregulated in high-risk HPV-positive cervical cancer, as its function involved the reduction of cancer growth via the suppression of glucose-6-phosphate dehydrogenase (*G6PD*) expression [26]. Similarly, high-risk HPV-mediated downregulation of hsa-miR-139-3p was suggested to promote oncogenesis in both cervical and head-and-neck cancer [27]. miR-155 was found to be a potential biomarker for HPV-positive cervical cancer, while induction of miR-20b overexpression by high-risk HPV promoted cervical cancer cell migration and invasion [28,29].

Due to its causal role in certain types of cancer, much of the focus in HPV research has been on high-risk HPV. As a result, little is known about the miRNA profiles of low-risk HPV infection involving common non-genital warts. Therefore, the purpose of the present study is to understand and identify the differences in miRNA expression between HPV-induced common warts and healthy skin in humans.

## 2. Methods

### 2.1. Sample Collection

After giving written informed consent, participants were recruited from the general population at King Abdullah University Hospital in Irbid, Jordan. Inclusion criteria involved being male and presenting with common warts, and patients were excluded if they suffered from any comorbid autoimmune or dermatological diseases. Paired samples of warts and healthy skin were obtained by means of shave biopsy from twelve participants, resulting in 12 wart samples and 12 normal skin samples (*n* = 24). Clinical diagnosis and procedures were performed by a resident dermatologist. All excised tissue samples were stored in RNA-SafeGuard reagent (GMbiolab Co., Ltd., Taiwan) and refrigerated at −20 °C. The present study was approved by the Institutional Review Board (IRB) at Jordan University of Science and Technology (JUST) (Ref. 19/105/2017).

### 2.2. RNA Isolation

RNeasy Mini Kit (Qiagen, Germany) was used to extract total RNA from the 24 tissue samples, and optional on-column DNase digestion was carried out. The quality and quantity of RNA were determined on BioTek’s PowerWave XS2 Spectrophotometer (BioTek Instruments, Inc., Winooski, VT, USA) while RNA integrity numbers were determined by means of the Agilent Bioanalyzer (Agilent Technologies, Inc., Santa Clara, CA, USA).

### 2.3. RNA Sequencing (RNA-seq)

Samples that met standards for quality control were shipped on dry ice to the Australian Genome Research Facility in Melbourne, Australia, where RNA-seq was performed on the Illumina HiSeq 2500 according to the manufacturer’s protocol. Real-time image analysis was carried out using the HiSeq Control Software (HCS) v2.2.68 (Illumina, Inc., San Diego, CA, USA) and Real Time Analysis (RTA) v1.18.66.3 (Illumina, Inc., San Diego, CA, USA). Afterwards, sequence data was generated through the Illumina bcl2fastq 2.20.0.422 pipeline (Illumina, Inc., San Diego, CA, USA).

### 2.4. Identification of Differentially Expressed miRNAs

Differentially expressed (DE) miRNAs were identified using the normalized sequence reads from all 24 samples. An miRNA was determined to be DE between warts and normal tissue samples if it had an adjusted *p*-value (AP) of less than or equal to 0.05.

### 2.5. Bioinformatic Analyses

The potential target genes of the identified miRNAs were predicted using miRDB (http://www.mirdb.org/), miRTarBase (http://mirtarbase.mbc.nctu.edu.tw/php/index.php), miRTargetLink Human (https://www.ccb.uni-saarland.de/), and TargetScan (http://www.targetscan.org/). miRNet (https://www.mirnet.ca/) was used to analyze the association between the list of miRNAs and disease. Gene ontology (GO) enrichment analysis and KEGG pathway enrichment of the potential miRNA target genes was carried out using miRTarBase and DAVID (http://david.abcc.ncifcrf.gov/). The GO terms included three criteria: molecular function (MF), cellular component (CC), and biological process (BP). The cutoff threshold was set at 0.05.

## 3. Results

### 3.1. Differentially Expressed miRNAs

Among the 1830 miRNAs included in the analysis, a total of 6 differentially expressed (DE) miRNAs were found in warts. Of these, 5 miRNAs were found to be upregulated and 1 miRNA was downregulated in warts compared to normal skin (Table 1). The fold change for each of the 12 control and 12 wart samples was individually plotted for the 6 DE miRNAs, and an overall consistent DE of the latter was shown in the wart samples (Appendix A).

### 3.2. Bioinformatics Analysis

The 6 candidate miRNAs were subject to further analysis to better understand their physiological functions. The potential target genes and associated diseases of the candidate miRNAs were predicted using the following web-based tools: miRDB, miRTarBase, miRNet, miRTargetLink Human, and TargetScan.

By using miRDB, a total of 737 predicted targets were found for hsa-miR-612, with 472 predicted targets for hsa-miR-647, 292 predicted targets for hsa-miR-3654, 338 predicted targets for hsa-miR-1914-3p, 213 predicted targets for hsa-miR-24-1-5p, and 1497 predicted targets for hsa-miR-27b-3p. By contrast, the use of miRTarBase revealed a predicted target number of 258 genes for hsa-mir-612, 58 genes for hsa-mir-647, 34 genes for hsa-mir-3654, 88 genes for hsa-mir-1914-5p, 141 genes for hsa-mir-1914-3p, 37 genes for hsa-mir-24-1, and 422 genes for hsa-mir-27b. Further, potential target genes were predicted using TargetScan, it revealed a total of 31 genes for hsa-mir-612, 4469 genes for hsa-mir-647, 3000 genes for hsa-mir-3654, 3946 genes for hsa-mir-1914, 1392 genes for hsa-mir-24-1, and 2737 genes for hsa-mir-27b.

Interaction networks for each candidate miRNA and overlapping target genes were created using miRTargetLink Human. Interactions with strong experimental evidence are shown in Figure 1. The term “strong evidence” indicates that the interaction of miRNAs with the target genes is supported by strong experimental methods from the literature, such as reporter gene assays. By contrast, “weak evidence” indicates that the interactions are supported by weaker experimental methods like microarrays. Information about the target genes was extracted from miRTarBase as well as from in-house generated data from miRTargetLink Human databases [30].

The target gene overlap between the 6 candidate miRNAs is shown in Figure 2 and Table 2. Three miRNAs, namely hsa-miR-27b, hsa-miR-1914-3p and hsa-miR-612, are at the center of the interaction network, having more than 12 interactions each. These three miRNAs have two common target genes (*OLR1* and *PLXND1*). Moreover, the results showed that two genes (*VEGFC* and *CREB1*) are likely to be potential targets of hsa-miR-27b, a finding that is supported by previous reports. The miRNet webtool was used to predict diseases associated with the candidate miRNAs, and it revealed a total of 33 associated diseases for hsa-miR-24-1, 29 diseases for hsa-mir-27b, and 2 diseases for hsa-mir-612. By contrast, no associated diseases were predicted for hsa-mir-647, hsa-mir-3654, and hsa-mir-1914 (Appendix A).

### 3.3. Functional Enrichment Analysis

GO enrichment analyses for miRNA target genes were performed using the DAVID online database webtool. The top 10 most significant GO terms of each criteria are presented in Figure 3, which shows that the target genes of the candidate miRNAs were mainly enriched in protein binding and poly(A) RNA binding on the MF level, enriched in nucleoplasm and nucleus on the CC level, and enriched in regulation of transcription, DNA-templated and transcription, DNA-templated on the BP level. The top 20 most significant KEGG pathway terms are presented. The miRNA target genes were mainly enriched in focal adhesion, ErbB signaling pathway, and adherens junction.

## 4. Discussion

In the present study, miR-27b was found to be upregulated in warts compared to normal skin, and this miRNA displayed the highest number of interactions (*n* = 27) with target genes. miR-27b, one of two miR-27 homologs, modulates adipocyte differentiation as well as adipogenesis regulation by targeting peroxisome proliferator-activated receptor gamma (*PPARγ*) [31]. High-risk HPV infection is reported to upregulate miR-27b levels in HPV-positive cervical cancer tissues which, in turn, increases cell proliferation and decreases apoptosis by inhibiting *PPARγ* expression [32,33,34]. By contrast, the inhibition of *PPARγ* expression was found to enhance the response of cervical cancer cells to radiation treatment, while ligand activation of the PPARγ nuclear receptor resulted in the induction of differentiation and apoptosis in non-small cell lung cancer cells [35,36].

In addition, strong evidence illustrated that miR-27b-3p targets the vascular endothelial growth factor C (*VEGFC*) and CAMP-responsive element binding protein 1 (*CREB1*) genes (Table 2). The VEGFC protein functions mainly in the process of lymphangiogenesis but has also been implicated in the promotion of tumor metastasis and growth [37]. *VEGFC* gene transcription was significantly increased in allergen-stimulated keratinocytes, but the VEGFC protein itself was not reported to directly affect the in vitro proliferation of epidermal keratinocytes [38,39]. In the context of high-risk HPV infection, *VEGFC* expression was stimulated by cigarette smoke and associated with the grade of cervical intraepithelial neoplasia [40,41]. On the other hand, the inhibition of CREB family members led to the reduction of papilloma formation in the murine epidermis via induction of apoptosis [42]. Correspondingly, *CREB1* knockdown was found to promote apoptosis in murine follicular cells [43]. Moreover, in bladder cancer cells, *CREB1* has been implicated in epithelial to mesenchymal transition, a carcinogenic process which is non-significantly associated with the HPV status of squamous cell oropharyngeal carcinoma [44,45].

miR-24-1 was found to be upregulated in warts compared to normal skin. miR-24 was reported to induce apoptosis and inhibit growth in laryngeal squamous cell carcinoma cells [46]. Moreover, keratinocyte differentiation normally promotes miR-24 expression, but this induction does not occur in keratinocytes infected with high-risk HPV [47]. Moreover, aberrant miR-24 expression was reported in HPV-positive cervical cancer cell lines [48]. Elevated miR-24 levels were associated with secreted frizzled-related protein 4 (*SFRP4*) levels in obese and diabetic subjects [49]. miR-24-1 was strongly upregulated during white adipocyte differentiation in a murine model, and its overexpression helped enhance adipocyte differentiation in murine mesenchymal stem cells [50,51].

Compared to normal skin, miR-3654, miR-647, and miR-1914 were found to be upregulated in warts. Both miR-647 and miR-1914 were found to be elevated in colorectal cancer tissues and cell lines as miR-647/1914 was found to promote the proliferation and migration of cancer cells by the downregulation of nuclear factor I/X [52]. By contrast, another study found that miR-1914 helped suppress the chemoresistant abilities of colorectal cancer cells also by nuclear factor I/X (*NFIX*) downregulation [53]. Moreover, miR-3654 has been identified for its involvement in prostate cancer progression [54].

Contrastingly, miR-612 was found to be downregulated in warts compared to normal skin. miR-612 plays a critical role in the development of colorectal cancer cells, and it was also found to have a suppressive role in reducing the stemness and tumor metastasis of hepatocellular carcinoma cells [55,56]. In hepatocellular carcinoma patients, miR-612 inhibits the metastasis and epithelial–mesenchymal transition of cancer cells by targeting the *AKT2* gene [57]. In addition, this miRNA has been reported as a potential regulator of *TP53* and *CD40* expression in adults with metabolic syndrome [58]. As can be seen, it appears that miR-612 possess antitumor functions in malignant lesions, which might explain why it was found to be downregulated in benign lesions such as warts.

With regard to interactions with other genes, miR-27b, miR-1914-3p, and miR-612 were found to have the highest number of interactions. The oxidized low-density lipoprotein receptor 1 (*OLR1*) and plexin D1 (*PLXND1*) genes were identified as common targets of the three miRNAs. The *ORL1* gene encodes for the lectin-type oxidized LDL receptor 1 (LOX-1), a protein that acts as the main oxidized LDL receptor on several different types of cells [59,60]. In the skin, the LOX-1 proteins on the surface of keratinocytes can be acted upon by nociception to produce leukotriene B(4), the latter of which induces an itch-associated response in a murine model [61]. LOX-1 also plays a role in the humoral immunity of the skin, as it is expressed on the surfaces of dermal dendritic cells [62,63]. On the other hand, the *PLXND1* gene is involved in body fat distribution and is abundantly expressed in tumor vasculature [64,65,66].

Despite the ubiquity of low-risk HPV infection, there is a dearth of information available regarding the expression profiles of non-genital warts. While miRNA profiles have been investigated in HPV-associated cancers, little is known about the changes in miRNA expression that occur in warts. As a result, the present study aimed to shed light on this issue and provide a much-needed comparison between the miRNA expression patterns of high- and low-risk HPV infection. One limitation of the present study was that all participants were male. However, as this is an exploratory study, only male subjects were included in order to reduce any gender-specific variation that might arise from sex-biased expression patterns.

## Figures and Tables

**Figure 1 biomolecules-09-00757-f001:**
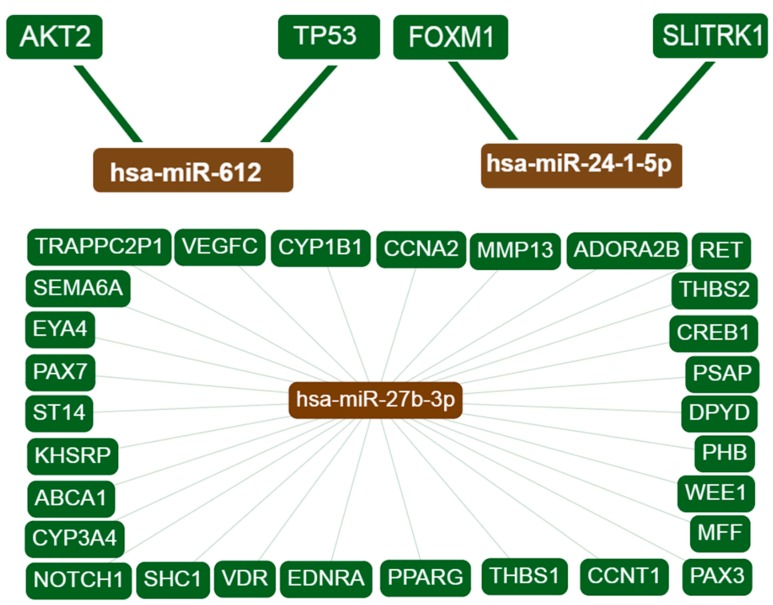
Interaction network of the target genes. Only interactions with strong evidence are shown. hsa-miR-27b-3p displayed 29 interactions while hsa-miR-612 and hsa-miR24-1 show 2 interactions each. hsa-miR-647, hsa-miR-3654, and hsa-miR-1914 showed no interactions with any genes.

**Figure 2 biomolecules-09-00757-f002:**
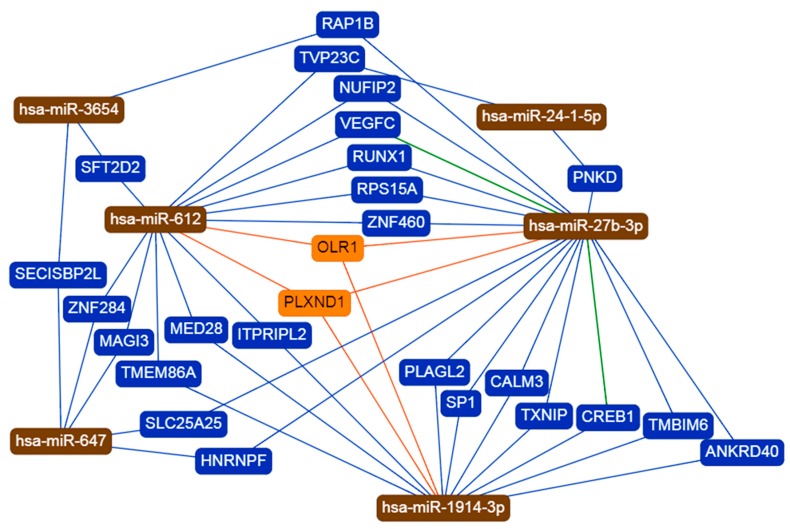
Target gene overlaps between 6 miRNAs. A total of 26 genes are targeted by two or more of the miRNAs. hsa-miR-612 showed 14 interactions, hsa-miR-647 shared 5 interactions, hsa-miR-3654 displayed 3 interactions, hsa-miR-1914-3p shared 12 interactions, hsa-miR-24-1-5p had 2 interactions, and hsa-miR-27b-3p showed 18 interactions. Blue and orange lines indicate weak evidence, while green lines showed strong evidence.

**Figure 3 biomolecules-09-00757-f003:**
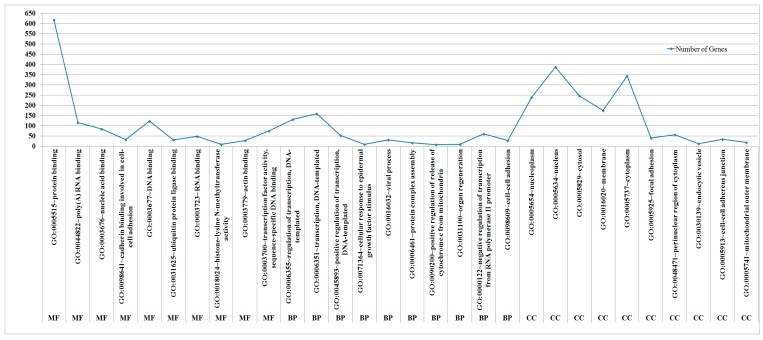
GO enrichment analyses illustrated the significantly enriched target genes of miRNAs in biological processes (BP), cellular component (CC), and molecular function (MF).

**Table 1 biomolecules-09-00757-t001:** Differentially expressed miRNAs associated with warts as sorted by adjusted *p*-value (AP).

miRNA Name	miRTarBase ID	FC	logCPM	QLF	*p*-Value	AP
miR27B	hsa-miR-27b	2.24	1.08	20.93	9.84 × 10^−5^	3.39 × 10^−4^
miR24-1	hsa-miR-24-1	1.87	1.49	17.29	2.98 × 10^−4^	8.89 × 10^−4^
miR3654	hsa-miR-3654	1.59	1.22	10.98	2.66 × 10^−3^	6.08 × 10^−3^
miR1914	hsa-miR-1914	1.65	0.65	8.79	6.32 × 10^−3^	1.30 × 10^−2^
miR612	hsa-miR-612	0.48	1.70	7.57	1.05 × 10^−2^	2.02 × 10^−2^
miR647	hsa-miR-647	1.51	0.53	6.50	1.69 × 10^−2^	3.07 × 10^−2^

FC: fold change; logCPM: base 2 logarithm of counts per million; QLF: quasi-likelihood F test; AP: adjusted *p*-value.

**Table 2 biomolecules-09-00757-t002:** List of overlapping target genes between the 6 candidate miRNAs.

miRNA Name	miRTarBase ID	Target Gene	Evidence Category	miRNA Name	miRTarBase ID	Target Gene	Evidence Category
hsa-miR-27b-3p	MIRT054253	*VEGFC*	Strong	hsa-miR-1914-3p	MIRT470812	*PLXND1*	Weak
hsa-miR-27b-3p	MIRT054313	*CREB1*	Strong	hsa-miR-1914-3p	MIRT481906	*ANKRD40*	Weak
hsa-miR-612	MIRT470846	*PLXND1*	Weak	hsa-miR-1914-3p	MIRT498181	*CREB1*	Weak
hsa-miR-612	MIRT471837	*NUFIP2*	Weak	hsa-miR-1914-3p	MIRT500628	*TXNIP*	Weak
hsa-miR-612	MIRT533683	*TMEM86A*	Weak	hsa-miR-1914-3p	MIRT510288	*MED28*	Weak
hsa-miR-612	MIRT534543	*RUNX1*	Weak	hsa-miR-1914-3p	MIRT513261	*CALM3*	Weak
hsa-miR-612	MIRT540989	*ZNF460*	Weak	hsa-miR-1914-3p	MIRT533677	*TMEM86A*	Weak
hsa-miR-612	MIRT542698	*RPS15A*	Weak	hsa-miR-1914-3p	MIRT561764	*PLAGL2*	Weak
hsa-miR-612	MIRT623347	*MAGI3*	Weak	hsa-miR-1914-3p	MIRT687811	*ITPRIPL2*	Weak
hsa-miR-612	MIRT631904	*VEGFC*	Weak	hsa-miR-24-1-5p	MIRT553243	*TVP23C*	Weak
hsa-miR-612	MIRT670847	*SFT2D2*	Weak	hsa-miR-24-1-5p	MIRT722489	*PNKD*	Weak
hsa-miR-612	MIRT674264	*ZNF284*	Weak	hsa-miR-27b-3p	MIRT046168	*RPS15A*	Weak
hsa-miR-612	MIRT675805	*MED28*	Weak	hsa-miR-27b-3p	MIRT046170	*CALM3*	Weak
hsa-miR-612	MIRT697940	*TVP23C*	Weak	hsa-miR-27b-3p	MIRT046236	*HNRNPF*	Weak
hsa-miR-612	MIRT702609	*ITPRIPL2*	Weak	hsa-miR-27b-3p	MIRT059147	*TXNIP*	Weak
hsa-miR-612	MIRT708482	*OLR1*	Weak	hsa-miR-27b-3p	MIRT065417	*TMBIM6*	Weak
hsa-miR-647	MIRT467849	*SLC25A25*	Weak	hsa-miR-27b-3p	MIRT400021	*PNKD*	Weak
hsa-miR-647	MIRT511456	*HNRNPF*	Weak	hsa-miR-27b-3p	MIRT469592	*RAP1B*	Weak
hsa-miR-647	MIRT565783	*SECISBP2L*	Weak	hsa-miR-27b-3p	MIRT471855	*NUFIP2*	Weak
hsa-miR-647	MIRT623341	*MAGI3*	Weak	hsa-miR-27b-3p	MIRT481919	*ANKRD40*	Weak
hsa-miR-647	MIRT674251	*ZNF284*	Weak	hsa-miR-27b-3p	MIRT500342	*ZNF460*	Weak
hsa-miR-3654	MIRT188227	*RAP1B*	Weak	hsa-miR-27b-3p	MIRT501601	*PLAGL2*	Weak
hsa-miR-3654	MIRT468655	*SECISBP2L*	Weak	hsa-miR-27b-3p	MIRT666295	*SLC25A25*	Weak
hsa-miR-3654	MIRT534359	*SFT2D2*	Weak	hsa-miR-27b-3p	MIRT682415	*OLR1*	Weak
hsa-miR-1914-3p	MIRT113263	*TMBIM6*	Weak	hsa-miR-27b-3p	MIRT686793	*SP1*	Weak
hsa-miR-1914-3p	MIRT451720	*OLR1*	Weak	hsa-miR-27b-3p	MIRT699900	*RUNX1*	Weak
hsa-miR-1914-3p	MIRT467369	*SP1*	Weak	hsa-miR-27b-3p	MIRT726920	*PLXND1*	Weak

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
