# Peer review of "Gene Expression Profiling of MicroRNAs in HPV-Induced Warts and Normal Skin"

_biomolecules, 2019, doi:10.3390/biom9120757_

Round 1

Reviewer 1 Report

Al-Eitan et al. present an interesting study concerning the miRNA expression profile in cutaneous HPV-induced warts. The Authors identify up- and down-regulated miRs and present suggestions of how these changes might contribute for the pathogenesis of warts. The study of micoRNAs in lesions induced by HPV is a very active field, but most works have focused on malignant lesions induced by high-risk types. The fact that the present paper focuses on cutaneous warts is critical for its novelty.

The work is well-structured, is appropriately designed and employs adequate methodologies. The results are clear and convincing and the conclusions are not overstated (in fact, they deserve to be made more explicit). The standards of English language are acceptable and I am only suggesting minor grammar adjustments (below).

Overall, the article contains significant novelty, and should be of interest for the readers of the Biomolecules. A few comments are listed below.

Abstract, line 20: "Strong evidence showed..." What evidence? Please be more specific saying what kind of analysis/results support this conclusion

Abstract, lines 21-23. Why are these genes important? Please add a short concluding sentence saying why these findings are important.

Introduction, lines 38-39. HPV are also grouped in genera based on L1 sequence homology. Please add a mention to this, otherwise it would seem that the HPV classification is only based on cancer risk and tissue tropism. Reference 10 is lacking the parentheses.

Methods. line 70. What was the reason for including only men? How will the patient's sex impact the analysis of results? This is a factor that needs to be addressed in the discussion section. Also, what were the exclusion criteria?

Methods, line 71. Please clarify: were these 12 normal and 12 pathological samples?

Methods, line 74-75. Did the study participants provide written informed consent?

Results, line 103. Please indicate how many miRs were actually included in this analysis, before pointing out the 6 that were differentially expressed.

Results, line 114. "...targets was found..." please correct to "targets were found".

Results, line 124 and others below. The Author mention "strong" experimental evidence and "weak" experimental evidence as criteria for including gene interactions in their analysis (e.g. again in Figure 1). The Authors need to explain what evidence do they have or where is it stored, how it was used, and what is the nature of this evidence.

The purpose of Table 3 is not clear - it mentions all kinds of conditions but the paper is about warts. I suggest withdrawing it or including only as supplementary material.

Discussion, lines 85-86. The significance of this function for the pathogenesis of cutaneous warts is unclear. Please discuss in greater depth or withdraw this sentence. The next sentence seems to bring a lot more "functional relevance" and it would be useful to discuss the mechanisms whereby CREB1 may regulate apoptosis.  

Discussion, lines 196-202. These contrasting observations deserve to be better discussed. Do the Authors think that the downregulation of these miRs is promoting the development of warts or countering it?

Discussion, lines 203-209. The same applies to this paragraph. What role do the Authors hypothesize for this miR in warts? It seems to have anti-tumor functions in other kinds of lesions.

Discussion. At some point, the Authors need to discuss the fact that their study only included male patients, and how this may reflect on their findings.

Author Response

Dear Ms. Xu,

I would like to extend my deepest thanks to the reviewer for the constructive comments with regard to the manuscript titled “Gene expression profiling of microRNAs in HPV-induced warts and normal skin”. I am pleased to submit the revised version of the paper that addresses each comment noted by the reviewers. The manuscript was also reviewed by a native language speaker in order to enhance its flow and scientific communication.

Comments by Reviewer 2

Al-Eitan et al. present an interesting study concerning the miRNA expression profile in cutaneous HPV- induced warts. The Authors identify up- and down-regulated miRs and present suggestions of how these changes might contribute for the pathogenesis of warts. The study of micoRNAs in lesions induced by HPV is a very active field, but most works have focused on malignant lesions induced by high-risk types. The fact that the present paper focuses on cutaneous warts is critical for its novelty.

The work is well-structured, is appropriately designed and employs adequate methodologies. The results are clear and convincing and the conclusions are not overstated (in fact, they deserve to be made more explicit). The standards of English language are acceptable and I am only suggesting minor grammar adjustments (below).

Overall, the article contains significant novelty, and should be of interest for the readers of the Biomolecules. A few comments are listed below.

Abstract, line 20: "Strong evidence showed..." What evidence? Please be more specific saying what kind of analysis/results support this conclusion

Removed term “strong” and added more detail about evidence.

Abstract, lines 21-23. Why are these genes important? Please add a short concluding sentence saying why these findings are important.

Added a short concluding sentence about the importance of these genes.

Introduction, lines 38-39. HPV are also grouped in genera based on L1 sequence homology. Please add a mention to this, otherwise it would seem that the HPV classification is only based on cancer risk and tissue tropism. Reference 10 is lacking the parentheses.

Added a reference to HPV grouping by genera and corrected the parentheses for Reference 10.

line 70. What was the reason for including only men? How will the patient's sex impact the analysis of results? This is a factor that needs to be addressed in the discussion section. Also, what were the exclusion criteria?

Added impact of patient’s sex to discussion, and added exclusion criteria to methods.

Methods, line 71. Please clarify: were these 12 normal and 12 pathological samples?

Yes, the 24 samples consisted of 12 normal and 12 pathological samples. This clarification was added to the text.

Methods, line 74-75. Did the study participants provide written informed consent?

All patients gave written informed consent as specified in the “Sample Collection” sub-section of the “Methods” section.

Results, line 103. Please indicate how many miRs were actually included in this analysis, before pointing out the 6 that were differentially expressed.

Added the following to section 3.1 of the results: “Among the 1,830 miRNAs included in the analysis, a total of 6 differentially expressed (DE) miRNAs were found in warts”

Results, line 114. "...targets was found..." please correct to "targets were found".

Corrected.

Results, line 124 and others below. The Author mention "strong" experimental evidence and "weak" experimental evidence as criteria for including gene interactions in their analysis (e.g. again in Figure 1). The Authors need to explain what evidence do they have or where is it stored, how it was used, and what is the nature of this evidence.

The terms “strong” and “weak” evidence were explained in the Results section 3.2, and an in-text citation was included for more details about the source of this evidence.

The purpose of Table 3 is not clear - it mentions all kinds of conditions but the paper is about warts. I suggest withdrawing it or including only as supplementary material.

Moved Table 3 to the Supplementary section.

Discussion, lines 85-86. The significance of this function for the pathogenesis of cutaneous warts is unclear. Please discuss in greater depth or withdraw this sentence. The next sentence seems to bring a lot more "functional relevance" and it would be useful to discuss the mechanisms whereby CREB1 may regulate apoptosis.

Withdrew sentence and discussed in greater detail the mechanisms by which CREB1 regulates apoptosis.

Discussion, lines 196-202. These contrasting observations deserve to be better discussed. Do the Authors think that the downregulation of these miRs is promoting the development of warts or countering it?

As there is still not enough evidence in the literature, it is still too early to state whether these miRNAs might promote or counter wart development.

Discussion, lines 203-209. The same applies to this paragraph. What role do the Authors hypothesize for this miR in warts? It seems to have anti-tumor functions in other kinds of lesions.

Discussed contrasting observations in greater detail.

At some point, the Authors need to discuss the fact that their study only included male patients, and how this may reflect on their findings.

Added impact of patient’s sex to discussion and how it may reflect on our findings.

Please do not hesitate to contact me if you need any additional information. I highly look forward to hearing from you.

Yours sincerely,

Dr. Laith N. AL-Eitan, MSc, PhD
Associate Professor of Human Genetics and Pharmacogenetics 

Department of Biotechnology & Genetic Engineering  
Faculty of Science and Arts
Jordan University of Science and Technology,
P.O.Box 3030, Irbid 22110, JORDAN
Email: lneitan@just.edu.jo
Tel.: +962-2-7201000 ext.: 23464 

Reviewer 2 Report

El-Aitan et al present miRNA expression profiles of low-risk HPV warts compared to healthy skin of 24 male patients. Six statistically significant DE miRNAs were identified and various in silico analyses were performed (interaction network, functional enrichment, disease association, KEGG pathway) in an effort to understand the underlying biology. Overall the work is extremely descriptive, there is confusion/discrepancy between the miRNA fold change values in table 1 and the text, and no conclusions can be made regarding any biological implications.

Major concerns

1) In section 3.1 Results, one miRNA was described as being highly expressed in warts compared to normal skin and five miRNAs as having lower expression in warts compared to normal skin. This description doesn't make sense with data in Table 1- where five miRNAs (27b, 24-1, 3654, 1914, 647) have fold changes between 1.5 and 2.2 (upregulated in warts compared to control) and one miRNA (612) has a fold change of 0.48 (downregulated in warts compared to control). This seems backwards.

2) miR27b is described as being "strongly downregulated" in warts (line 169), contrary to the FC data in Table 1. Even if these terms (or positive/negative signs) were switched, the fold change is only 1.16 on a log base 2 (=2.2-fold). I wouldn't describe this as "strongly" up- or down-regulated

3) All connections regarding biological implications are via in silico analysis, which merges datasets from many different tissues/disease states/biological contexts, thus only highly speculative possibilities can be mentioned, no conclusions can really be drawn.

4) The study was done using male subjects only. Data from females should be included.

5) Data should be broken down by subject as well- were any particular miRNAs consistently up or down in all subjects? If not then what fraction of subjects displayed significant FC? The FC data could be plotted in a way to indicate this

Author Response

Dear Ms. Xu,

I would like to extend my deepest thanks to the reviewer for the constructive comments with regard to the manuscript titled “Gene expression profiling of microRNAs in HPV-induced warts and normal skin”. I am pleased to submit the revised version of the paper that addresses each comment noted by the reviewer. The manuscript was also reviewed by a native language speaker in order to enhance its flow and scientific communication.

Comments by Reviewer 1

El-Aitan et al present miRNA expression profiles of low-risk HPV warts compared to healthy skin of 24 male patients. Six statistically significant DE miRNAs were identified and various in silico analyses were performed (interaction network, functional enrichment, disease association, KEGG pathway) in an effort to understand the underlying biology. Overall the work is extremely descriptive, there is confusion/discrepancy between the miRNA fold change values in table 1 and the text, and no conclusions can be made regarding any biological implications.

Major concerns

In section 3.1 Results, one miRNA was described as being highly expressed in warts compared to normal skin and five miRNAs as having lower expression in warts compared to normal skin. This description doesn't make sense with data in Table 1- where five miRNAs (27b, 24-1, 3654, 1914, 647) have fold changes between 1.5 and 2.2 (upregulated in warts compared to control) and one miRNA (612) has a fold change of 0.48 (downregulated in warts compared to control). This seems backwards.

Corrected the aforementioned issue to include the following: 5 miRNA were found to be upregulated and 1 miRNA was downregulated in warts compared to normal skin (Table 1). Also, logFC values were replaced with FC for a better visual clarification as per reviewer suggestions (Table 1).

miR27b is described as being "strongly downregulated" in warts (line 169), contrary to the FC data in Table 1. Even if these terms (or positive/negative signs) were switched, the fold change is only 1.16 on a log base 2 (=2.2-fold). I wouldn't describe this as "strongly" up- or down-regulated

Corrected terminology (upregulated instead of downregulated) and removed the “strongly” description.

All connections regarding biological implications are via in silico analysis, which merges datasets from many different tissues/disease states/biological contexts, thus only highly speculative possibilities can be mentioned, no conclusions can really be drawn.

All conclusions drawn from the in silico analysis were omitted.

The study was done using male subjects only. Data from females should be included.

As this is an exploratory study, only male subjects were included in order to reduce any gender-specific variation that might arise from sex-biased expression patterns. This limitation was included in the Discussion.

Data should be broken down by subject as well- were any particular miRNAs consistently up or down in all subjects? If not then what fraction of subjects displayed significant FC? The FC data could be plotted in a way to indicate this

The fold change for individual subjects was plotted for the 6 miRNAs (Supplementary figure S1) and mentioned in sub-section 3.1 of “Results”.

Please do not hesitate to contact me if you need any additional information. I highly look forward to hearing from you.

Yours sincerely,

Dr. Laith N. AL-Eitan, MSc, PhD
Associate Professor of Human Genetics and Pharmacogenetics 

Department of Biotechnology & Genetic Engineering  
Faculty of Science and Arts
Jordan University of Science and Technology,
P.O.Box 3030, Irbid 22110, JORDAN
Email: lneitan@just.edu.jo
Tel.: +962-2-7201000 ext.: 23464 

Round 2

Reviewer 2 Report

revised version satisfied my prior concerns